# Suppressive Effect of Soil Microbiomes Associated with Tropical Fruit Trees on *Meloidogyne enterolobii*

**DOI:** 10.3390/microorganisms10050894

**Published:** 2022-04-25

**Authors:** Milad Rashidifard, Hendrika Fourie, Samad Ashrafi, Gerhard Engelbrecht, Ahmed Elhady, Mieke Daneel, Sarina Claassens

**Affiliations:** 1Unit for Environmental Sciences and Management, North-West University, Private Bag X6001, Potchefstroom 2531, South Africa; driekie.fourie@nwu.ac.za (H.F.); gerhardengelbrecht38@gmail.com (G.E.); mieke@arc.agric.za (M.D.); sarina.claassens@nwu.ac.za (S.C.); 2Institute for Epidemiology and Pathogen Diagnostics, Federal Research Centre for Cultivated Plants Julius Kühn-Institut, 38126 Braunschweig, Germany; samad.ashrafi@julius-kuehn.de (S.A.); ahmed.elhady@julius-kuehn.de (A.E.); 3Agricultural Research Council-Tropical and Subtropical Crops (ARC-TSC), Private Bag X11208, Mbombela 1200, South Africa

**Keywords:** bacteria, biocontrol, fungi, plant-parasitic nematode, root-knot nematode, soil microbiome

## Abstract

Plant-parasitic nematodes are one of the main biotic factors limiting agricultural production worldwide, with root-knot nematodes (*Meloidogyne* spp.) being the most damaging group. This study was conducted to evaluate the efficacy of soil microbiomes, associated with various subtropical fruit trees, on the management of a *Meloidogyne enterolobii* population. Of 14 soil microbiomes tested for nematode suppression, 9 samples in the first experiment and 10 samples in the repeat experiment had significantly (*p* ≤ 0.05) lower numbers of eggs and J2 compared to the untreated control. The highest nematode suppression was recorded for SA12 extracted from a papaya orchard with a 38% reduction in the nematode population density. In addition, the presence of some bacteria (*Bacillus aryabhattai*, *B. funiculus* and *B. simplex*) and fungi (*Metarhizium marquandii*, *Acremonium* sp. and *Mortierella* sp.) was correlated to a higher suppression potential in some samples. Substantial variations were observed for the diversity of bacterial and fungal isolates among the samples collected from various crop hosts and regions. This suggests that the nematode suppression potential of different soil microbiomes highly depends on the abundance and diversity of fungal and bacterial strains present in the soil. The study confirmed that among all variables, soil dryness, pH, Fe, Zn, organic matter, altitude, and crop cultivar strongly influenced the soil microbial composition.

## 1. Introduction

The human population is estimated to reach 9.7 billion by 2050, thus requiring 70% more food to be available for human consumption [1]. Subtropical and tropical crops in subtropical regions of South Africa, including avocado (*Persea americanum*), banana (*Musa* sp.), several citrus cultivars (*Citrus* spp.), guava (*Psidium guajava*), litchi (*Litchi chinensis*), mango (*Mangifera indica*), macadamia (*Macadamia integrifolia*) and papaya (*Carica papaya*), make up most of the agricultural production in this region [2,3,4]. These high-value crops are rich in proteins, carbohydrates, vitamins and beneficial oils and play an increasingly important role in the modern diets of humans (ARC–Institute for Tropical and Subtropical Crops, 2003). However, factors such as nutrient shortages and plant pathogen infections, especially plant-parasitic nematodes (PPN), can severely limit food production and, as a result, reduce food security [5]. Plant-parasitic nematodes (PPN) are major agricultural pests worldwide [5], among which, root-knot nematodes (RKN) are considered as one of the main factors that severely limit crop production [6]. Recent reports from South Africa show these nematodes parasitize various grains [7], vegetables [8,9] and fruit crops [3,10]. Furthermore, the geographical niches of some economically important nematode pest species are rapidly extending [9].

To control PPNs and alleviate their destructive effect on food security, chemical nematicides were predominantly utilized for nematode management. However, due to their risks to human, animal and environmental health, the use of these compounds has either been significantly reduced or banned. This has led to increased interest in research regarding more sustainable management strategies including microbially derived biological control approaches. Numerous bacteria and fungi have been identified with various modes of action contributing to nematicidal potential [11,12,13]. For example, the representatives of bacterial genera, including *Bacillus*, *Streptomyces*, *Pasteuria* and *Pseudomonas*, have nematicidal properties with mechanisms of action ranging from direct parasitism to the production of secondary metabolites that exhibit nematicidal characteristics [11,14,15]. Fungal species such as *Pochonia chlamydosporia* and *Purpureocillium lilacinum* from different families are also known to parasitize PPNs either directly and/or by producing nematicidal compounds [16,17,18].

Recently, more studies aimed to identify the nematophagous fungi and bacteria or a consortium of them that can disrupt the biological and/or physiological processes of PPNs [14]. The use of culture-independent technologies, such as next-generation sequencing (NGS) and microbial community fingerprinting, have provided valuable insights into identifying microbial consortia with nematicidal potential [19]. The application of these technologies has been used to identify numerous microbes that are associated with reducing nematode population densities. These include representatives of the bacterial genera *Acinetobacter*, *Arthrobacter*, *Bacillus*, *Brevundimonas*, *Chryseobacterium*, *Flavobacterium*, *Flexibacter*, *Lysobacter*, *Methylobacterium*, *Microbacterium*, *Micrococcus*, *Pasteuria*, *Pseudomonas*, *Rhizobium*, *Streptomyces*, *Steroidobacter*, *Sphingopyxis* and *Variovorax* [19,20,21]. Examples of fungi identified in PPN-suppressive soils include: *Arthrobotrys*, *Catenaria*, *Cladosporium*, *Dactylella*, *Drechslerella*, *Fusarium*, *Haptocillium*, *Hirsutella*, *Mortierella*, *Nematophthora*, *Pochonia*, *Preussia*, *Purpureocillium*, *Stachybotrys* and *Trichoderma* [14,18,19,22].

In agricultural soil, plant–microbe interactions within the rhizosphere are significantly impacted by, amongst others, management practices such as tillage and crop genotype, which alter the micro-environment [23,24,25]. Various subtropical and tropical trees are long-term, perennial plants in which soil microbial communities could become well established over time and benefit the host. This is in contrast with row crops, where their geographical niches are disrupted on a seasonal basis when crops are removed after harvesting. According to the phenotypical data from previous surveys, nematode damage has not been reported before in orchards where especially avocado, mango and macadamia were grown in the Burgershall, Malalane and Nelspruit areas, Mpumalanga province, South Africa [2]. Therefore, this study aimed to evaluate the nematode control potential of the soil microbiomes associated with the rhizospheres of various perennial subtropical tree crops on *Meloidogyne enterolobii*, widely spread species in subtropical regions of South Africa [9], under greenhouse conditions. It was hypothesized that the soil microbiomes (microbial communities) associated with subtropical tree crops could have the potential to reduce the damage caused by PPN.

## 2. Materials and Methods

### 2.1. Soil Sampling and Extraction of Microbial Communities from the Soil

The soil samples were collected from the rhizospheres of fruit trees in 14 orchards in three regions around Nelspruit in the Mpumalanga province, South Africa (Table 1). From each orchard, two rhizosphere subsamples were obtained from five trees using a shovel at a depth range of 15 to 35 cm (from surface). The collected soil samples were pooled into one composite sample per orchard. Samples were then stored in a cold room (6 °C) after being transferred to the Nematology Laboratory at the North-West University, Potchefstroom, North-West province, South Africa. Six subsamples (100 g each) were taken from each of the fourteen composite soil samples (one per orchard) and subjected to microbial extraction following the protocol of Elhady et al. [26]. Briefly, each sample was thoroughly mixed with 100 mL of 0.9% sterile saline water for 30 min in a shaker. Samples were left for 60 min for soil particles to settle down. The supernatant of each sample was decanted into a 50 mL capacity plastic tube and centrifuged at 500× *g* for 2 min to pellet the content, after which, the supernatant of each sample was again transferred into separate sterile 50 mL capacity tubes. The 14 tubes, each containing the supernatant with the microbial communities from one orchard’s soil extracts, were then centrifuged at 4000× *g* for 20 min for the microbial communities to be pelleted. The microbial pellet in each tube was resuspended by adding 50 mL of double distilled water (ddH_2_O) and shaking it for 30 s.

### 2.2. Soil Chemical and Physical Properties Measurement

Soil samples were analyzed by the Agricultural Research Council–Tropical and Subtropical Crops (ARC-TSC) soil analytical laboratory (Nelspruit). A representative sample per orchard was selected for physico-chemical analyses (pH (water), Res (Ohms), P (Bray 1) (mg/kg), Na (mg/kg), Mg (mg/kg), K (mg/kg), Ca (mg/kg), Al (mg/kg), Zn (mg/kg), Cu (mg/kg), Mn (mg/kg), Fe (mg/kg), Ca/Mg, Ca + Mg/K, %organic matter, %C, %sand, %silt and %clay). The soil was dried in the oven at 40 °C, subsequently crushed and passed through a 2 mm aperture sieve and mixed thoroughly. Thereafter, soil physical and chemical properties were determined according to standard methods (HTTS, 1990) (Appendix A).

### 2.3. Preparation of the Experiments and Nematode Inoculation

Tomato seeds of the RKN-susceptible cultivar Moneymaker [27] were surface-sterilized for 15 min in 1.5% sodium hypochlorite, rinsed twice using ddH_2_O and sowed in a tray containing sterile Culterra Compost (30 dm^3^) (http://culterra.co.za/category/products/soil/, accessed on 11 November 2021). One week prior to the experiment, plastic pots (2 L capacity) were filled with an autoclaved mixture of sandy soil (96% sand) and dry compost (ratio 2:1 *v*/*v*). The 4-leaf stage tomato seedlings that were grown from seeds in the compost mixture were individually transplanted into individual pots (representing one tomato plant per pot) containing the soil:compost substrate. Next, each of the resuspended pellets (50 mL) of the extracted microbial communities extracted from the 14 localities sampled were individually inoculated onto the root systems of each of the transplanted tomato seedlings. An untreated control was also prepared by adding 50 mL of ddH_2_O to the roots of an additional tomato seedling (as was done with the microbial soil extracts), totaling 15 treatments.

The eggs and second-stage juveniles (J2) of a previously identified population of *M. enterolobii* [9], which was reared in the roots of the susceptible tomato cultivar Moneymaker for 60 days, were extracted using an adapted NaOCl extraction method [28]. The total number of eggs and J2 were counted using a De Grisse counting dish [29] and a Nikon SMZ 1500 dissection microscope (60× magnification). Two weeks after inoculation of the 15 microbial community extract treatments, the roots of each tomato seedling were exposed by removing the soil around it and were inoculated by approximately 2000 eggs and J2s, with the removed soil being replaced after nematode inoculation. The potted tomato plants, inoculated with both microbial communities and RKN, were maintained in a glasshouse at an ambient temperature range of 22–28 ± 1.5 °C and a photoperiod of 14L:10D and watered at least three times a week with borehole water (pH (H_2_O) = 7.3). Nutrients were provided to the tomato plants by adding 50 mL (2 g/L) of a Starke Ayres Nutrifeed solution (nitrogen 6.5%; phosphorous 2.7%; potassium 13.0%; calcium 7.0%; magnesium 2.2%; sulphur 7.5%) to each pot every other week according to soil analysis. Seven weeks (49 days) after nematode inoculation, which provided enough time for the nematodes to complete at least two generations, the experiment was terminated. The experiment was repeated under the same conditions except for the ambient temperature range that was higher over the 49-days period, namely 22–32 ± 1.5 °C, to validate the data from the first experiment.

### 2.4. Nematode Extraction and Evaluation

The plants were uprooted and prepared for extraction and the determination of the egg and J2 population densities from the root system of each tomato plant. This was performed by excising the roots of each plant from its aerial plant parts and washing the roots using a gentle stream of tap water to remove excess debris. After this, the roots of each plant were weighed and then cut into 0.5 cm pieces before submitting them to the adapted NaOCl method [28], which was used to extract the eggs and J2. The egg and J2 numbers per root system were determined as explained above to obtain the inoculum [29]. In addition, the Oostebrink’s reproduction (Rf) value for each of the microbial community extract treatments and the untreated control were determined, where Rf = final population (Pf)/initial population (Pi) [30]. The suppression effect of each microbial community extract treatment on the RKN population was also calculated based on the following formula:
$\left(\frac{X\times 100}{Y}\right)-100$, where “*X*” refers to the final number of eggs and J2 (Pf) and “*Y*” is the initial nematode population (Pi) [31].

### 2.5. Statistical Analyses

A randomized complete block design (RCBD) with six replicates (one plant per pot; six pots for each treatment) was selected for the experimental layout. Nematode data for the eggs and J2 counts per root system was log(x) transformed as the data were not normally distributed. The nematode data as well as data for selected abiotic parameters (pH, K, Zn, Fe, Al and clay%) and altitude (environmental parameter) were subjected to one-way ANOVAs (Statistica, Version 13.3). The Tukey’s HSD Test (*p* ≤ 0.05) was performed to separate the means of the nematode and abiotic data for the 15 treatments (including the untreated control) included in each experiment. Data from both experiments were then analyzed using Factorial Analysis of Variance (ANOVA) (Statistica, Version 13.3, TIBCO Software Inc., Palo Alto, CA, USA) with time (the two experiments) being the main factor and treatments the sub-factor. The graphs presented in the results were generated using GraphPad Prism version 8.02.

### 2.6. DNA Extraction, Next-Generation Sequencing and Sequence Analysis

The DNA was extracted from the same soil sources of each of the 14 sites used for the microbe–nematode greenhouse evaluation. For DNA extraction, the Qiagen DNeasy PowerMax Soil Kit was used for 10 g of soil obtained from each orchard by following the user manual provided by the manufacturer. The V3–V4 regions of 16S rRNA genes were amplified using the primers 341F [32] and 806R [33]. Fungal ITS fragments were amplified by gITS7 [34] and ITS4 primers [35]. PCR amplifications were performed as described in Fernandez-Gnecco et al. [36]. The DNA fragments’ length and quality were detected on 2% agarose gel during electrophoresis. Samples with a visible band between 400 and 450 bp were chosen for further analysis. The PCR products were mixed at equal density ratios and were purified with the Qiagen Gel Extraction Kit (Qiagen, Hilden, Germany).

The libraries were generated using the NEBNext^®^ UltraTM DNA Library Prep Kit for Illumina and quantified using Qubit and qPCR and analyzed by using the Illumina platform. Amplicon sequencing of the ITS2 or 16S rRNA genes was carried out using 2 × 250 bp paired-end high-throughput sequencing on an Illumina HiSeq 2500 platform by Novogene (Cambridge, UK).

### 2.7. Sequencing Data Processing, OTU Cluster and Taxonomic Annotation

Paired-end reads were assigned to samples based on their unique barcodes and truncated by cutting off the barcode and primer sequences. Paired-end reads were merged using FLASH (V1.2.7) [37]. Quality filtering on the raw tags was performed under specific filtering conditions to obtain the high-quality clean tags [38] according to the Qiime (V1.7.0) [39]. The tags were compared with the reference database (Gold database; http://drive5.com/uchime/uchime_download.html, accessed on 5 August 2021) using the UCHIME algorithm (UCHIME Algorithm) [40]. Finally, to obtain the effective tags, the Chimera sequences were removed [41]. Sequence analyses were performed using Uparse software (v7.0.1001) according to [42], using all the effective tags. Sequences with ≥97% similarity were assigned to the same OTUs. A representative sequence for each OTU was screened for further annotation. Annotation of bacterial and fungal species at each taxonomic rank was performed using the mothur software against the SSUrRNA database of SILVA [43]. Finally, the OTUs’ abundance information was normalized using a standard sequence number corresponding to the sample with the least sequences.

### 2.8. Data Quality Control, Operational Taxonomic Unit Identification and Alpha and Beta Diversity

A total of 14 samples were presented in both bacteria (16S) and fungi (ITS) datasets. Data filtering was performed using Microbiome Analyst [44] based on default parameters and total normalization. All features with a minimum count below four and prevalence lower than 20% in the samples were excluded from the data as well as those with variance percentages lower than 10 based on the inter-quantile range (IQR). Alpha and beta diversities were conducted using the phyloseq package [45]. The Shannon index was used to show the diversity level in the samples (alpha diversity), and T-test/ANOVA was used as a statistical method. Nonmetric multidimensional scaling (NMDS) diagrams were used to show the differences between the various rhizosphere microbial communities (beta diversity). At the same time, statistical significance was found using permutational MANOVA (PERMANOVA) in Microbiome Analyst [44] and the Bray–Curtis dissimilarity distance to determine differences and similarities.

### 2.9. Linear Discriminant Analysis (LDA) Effect Size (LEfSe) and Heatmap Clustering

The Linear Discriminant Analysis (LDA) Effect Size (LEfSe) [46] was used based on a non-parametric factorial Kruskal–Wallis (KW) sum-rank test to identify features with significant differential abundance with regard to experimental factor (region), followed by Linear Discriminant Analysis (LDA) to calculate the effect size of each differentially abundant feature. Features were considered to be significant based on their adjusted *p*-value of 0.05. Hierarchical clustering was performed with the *hclust* function in package *stat* implemented in Microbiome Analyst [44] and shown as heatmaps and dendrograms for the 20 most abundant bacterial and fungal taxa.

## 3. Results

### 3.1. Number of Eggs and J2 per Plant

A significant interaction (*p* = 0.000; *F* = 5.43) was recorded for the numbers of eggs and J2 per root system between the two experiments due to the significant differences being evident for SA2, SA3 and SA12 treatments between the two experiments (Figure 1 and Appendix A). Nine of the fourteen microbial community treatments (SA1: banana, SA2: papaya, SA3: mango, SA6: macadamia, SA7: mango, SA8: citrus, SA9: avocado, SA12: papaya and SA14: banana) had significantly (*p* ≤ 0.05) lower numbers of eggs and J2s of *M. enterolobii* per tomato root system than that of the untreated control in the first experiment. Tomato roots inoculated with microbial communities from orchard SA12 (papaya) and SA8 (citrus) had the lowest number of eggs and J2 in the roots (1246 and 2056, respectively), while the untreated control had the highest (8382). The other five treatments (SA4: citrus, SA5: litchi, SA10: litchi, SA11: avocado and SA13: macadamia) showed no significant differences with regard to the number of eggs and J2 compared to that of the untreated control (*p* ≤ 0.05) (Figure 1).

In the repeat experiment, 10 of the 14 microbial communities had significantly (*p* ≤ 0.05) lower numbers of *M. enterolobii* eggs and J2s per root system compared to that of the untreated control (Figure 1 and Appendix A). The lowest number of eggs and J2 per root system in this experiment was recorded for SA12 (3270), followed by SA13 (3678). The untreated control had the highest number of eggs and J2 per root system in the second/repeat experiment (12,209). There were no significant differences (*p* ≤ 0.05) between four of the treatments (SA2, SA3, SA7 and SA10) compared to the untreated control (Figure 1).

### 3.2. Reproduction Factor (Rf) Value

There was a significant interaction (*p* = 0.000; *F* = 5.47) for the experiment x treatments due to the significant (*p* ≤ 0.05) differences of SA2, SA3 and the untreated control between the two experiments (Figure 2 and Appendix A). The Rf value ranged between 0.6 (SA12: papaya) and 4.2 (the untreated control) in the first experiment. During this experiment, except for three treatments (SA5: litchi; SA11: avocado; and SA13: macadamia), the other 11 microbial extract treatments had significantly lower Rf values compared to the untreated control (*p* ≤ 0.05) (Figure 2).

For the repeat experiment, the Rf value fluctuated between 1.6 (SA12: papaya) and 6.1 (the untreated control), with all 14 microbial community extract treatments having significantly (*p* ≤ 0.05) lower Rf values compared to the untreated control.

### 3.3. Suppression Effect of Microbial Communities on the Nematode Population

Among the 14 microbial communities extracted from rhizospheres of various fruit trees in the first experiment, only SA12 extracted from papaya prohibited the reproduction of *M. enterolobii*, showing the highest nematode suppression effect (Figure 3). This sample reduced the *M. enterolobii* Pi by 38% followed by SA8 (citrus), which only allowed a 3% increase in the nematode population. The other samples allowed variable reproduction, with the highest increase observed for the untreated control, which was 319% higher than the Pi. During the repeat experiment, the highest suppression was observed for SA12 (papaya), which only increased the Pi by 64%, followed by SA13 (macadamia), which allowed an 84% increase in the nematode population. Unlike the first experiment in the repeat experiment, none of the 14 microbial extracts could reduce the Pi, and the highest increase in the nematode population density was observed in the untreated control (510%).

Statistical analysis indicated that based on Tukey’s test, six selected soil chemical parameters as well as the altitude were significantly different (*p* ≤ 0.05) among the three geographic regions sampled (Tomahawk Citrus Malelane, ARC-TSC Nelspruit and ARC-TSC Burgershall) (Table 2), and these differences might have contributed towards impacting the soil microbial diversity.

### 3.4. Bacterial and Fungal Datasets

The bacterial NGS dataset (16S) had a total of 113,198 reads and 925 features or taxa. Among these, 283 low-abundance and 22 low-variance features were removed from the dataset based on prevalence and inter-quartile range (IQR). Hence, 195 features remained after the data filtering. The fungal NGS dataset (ITS) contained a total of 479,093 reads and 1438 features or taxa, from which, 424 low-abundance and 42 low-variance features were removed based on prevalence and IQR, respectively. Finally, 373 features remained after data filtering.

### 3.5. Alpha Diversity

Both Figure 4A,B and Figure 5A,B represent the alpha diversities among bacterial and fungal strains, respectively, based on their crop hosts and localities. The results showed that SA12 and SA4 from papaya and citrus rhizospheres, respectively, had the highest bacterial diversity (±4.18); however, SA6 extracted from macadamia rhizospheres had the lowest (±2.62) (Figure 4A). The alpha diversity biplots using the Shannon index categorized based on sample location indicated that the highest and lowest bacterial diversity was observed in the samples taken from ARC-TSC Burgershall (±3.42–4.18) and Tomahawk Citrus Malelane (±3.18–4.18), respectively (Figure 4B).

The alpha diversity analysis based on the Shannon index for individual samples showed that SA9 extracted from avocado rhizospheres (±4.4) and SA12 (±2.81) from papaya rhizospheres had the highest and lowest fungal diversity, respectively (Figure 5A). Moreover, ARC-TSC Nelspruit showed a more diverse fungal community (±3.10–4.40) than those of other locations based on alpha diversity biplots (Figure 5B).

### 3.6. Beta Diversity

The beta diversity diagram, illustrated using NMDS (Figure 6 and Figure 7), showed the differences between various samples regarding their microbial community compositions. Based on the beta diversity analysis using the bacterial dataset, the samples from SA1, SA2 and SA6 did not cluster with others, showing that rhizospheres of these orchards had different bacterial compositions. The other orchards grouped closer to the zero *X*- and *Y*-axis, suggesting they share similarities in their bacterial taxonomic profiles (Figure 6).

The NMDS based on the fungal dataset indicated the following orchards: SA2, SA6, SA7 and SA11 grouped separately, suggesting their different fungal taxonomic profiles compared to those of the other orchards. However, the other orchards clustered together close to zero on both the *X*- and *Y*-axis, meaning these orchards share a similar fungal composition in the soil (Figure 7).

### 3.7. Microbial Diversity

The taxonomic composition of the 20 most abundant bacterial taxa at the genus level (Appendix A) showed that *Bradyrhizobium* was the most abundant bacterial genus (25%) followed by unidentified Actinobacteria (18%), and *Actinoplanes* was the least abundant genus (3%). The diversity heatmap showed *Bacillus funiculus* was more abundant in the SA1 sample, *B. simplex* in SA9 and *B. aryabhattai* as well as *Staphylococcus haemolyticus* were more abundant in SA12 (Figure 8). Moreover, *Enterobacter kobei* and *Bradyrhizobium elkani* were the most abundant bacterial species in SA13.

Based on the taxonomic composition of the 20 most abundant fungal taxa at the genus level (Appendix A), *Apiotrichum* was the most abundant genus (19%), followed by *Gibberella* (12%) while *Talaromyces*, *Acremonium* and *Trichoderma* were the least abundant (3%). The heatmap of the fungal communities indicated that *Hypomyces* and *Metarhizium marquandii* were more frequent in the SA3 sample, *Mortierella* sp. in SA14 and *Acremonium* sp. in SA9. Furthermore, *Trichoderma rossicum* was found to be more abundant in SA5 and *Staphylotrichum coccosporum* in SA2. Ultimately, *Cladophialophora* sp. and *Talaromyces* sp. were more abundant in SA13 (Figure 9).

The LEfSe analyses indicated no significant differences (*p* ≤ 0.05) regarding the abundance of bacterial species between the three sampled regions where the 14 orchards were located. However, the same analyses for the fungal community showed that 33 species were significantly (*p* ≤ 0.05) different in terms of abundance concerning the three regions where sampling was carried out. Of the 33 species, 14 species were significantly more abundant (*p* ≤ 0.05) in ARC-TSC Burgershall, 10 species in ARC-TSC Nelspruit and 9 species in Tomahawk Citrus Malelane (Figure 10).

## 4. Discussion

During this study, significantly lower numbers of *M. enterolobii* eggs and J2 (64% and 71% of the samples in the first and repeat experiments, respectively) were recorded from tomato roots grown in soils that were inoculated with microbial communities extracted from 14 orchards’ rhizospheres. Additionally, a 38% reduction and only a 3% increase in the initial RKN inoculum population were recorded in the first experiment when microbial communities from papaya (SA12) and the citrus (SA8) rhizospheres were inoculated to the soil where tomato seedlings were grown, respectively. Although no reduction in the initial *M. enterolobii* egg and J2 inoculation numbers, which is similar to their resultant Rf values, were recorded for any of the microbial community extracts from the fruit tree rhizospheres for the second experiment, the final densities were substantially lower than those of the untreated control treatment, indicating their potential biocontrol efficacy. These results demonstrate the potential of microbial communities to be utilized in nematode management where chemical control is limited or not an option. This phenomenon was reported by a study carried out in the United States of America (Florida) where microbial communities reduced the reproduction ability of *M. incognita* in roots of pepper (*Capsicum annuum*) [47]. Furthermore, it was also reported that variation existed in the suppression effect of soil microbiomes associated with different plant species [48]. Other studies confirmed significantly lower reproduction rates of *M. hapla* in tomato roots grown in natural soil than those grown in sterilized soil, showing the essential role of the soil microbiome in plant health [49]. The numbers of eggs and J2 in the repeat experiment of our study were typically higher than those in the first experiment. This can be explained by a 4 °C higher ambient temperature difference for the repeat experiment, being 22–32 °C compared to 22–28 °C in the first experiment. *M. enterolobii* is a thermophilic species, which means they develop better in warmer climates.

The Shannon index’s alpha diversity analysis indicated a generally higher species abundance and evenness for bacterial communities compared to those of fungal communities (Figure 4 and Figure 5). This was evident for the soil microbiome of SA12 extracted from papaya rhizospheres that had the lowest fungal abundance but one of the highest bacterial abundances among the soil samples investigated. The higher tolerance of bacterial isolates to drought and dry conditions compared to those of fungal isolates might be a reason for this [50], as this sample was taken from a very dry orchard with no irrigation system. However, in contrast with this, another study showed that soil bacteria are more prone to drought than fungi [51]. Another possible reason for the observed difference might be the presence of an optimal pH range for bacterial growth in this particular site [52] (Appendix A).

The highest bacterial diversity was found in the samples from the ARC-TSC Burgershall region, while the lowest diversity was among the samples collected in the Tomahawk Citrus region (Figure 4B). There were significant differences (*p* ≤ 0.05) in terms of available Fe and Zn in the soil as well as the altitude between the two regions (Table 2). The average altitude for sampled orchards in the ARC-TSC Burgershall region was 770 m.a.s.l., compared to a substantially lower 299 m.a.s.l. for the orchards at the Tomahawk Citrus region. Previous studies have explicitly confirmed the effect of elevation on bacterial diversity [53,54]. Additionally, increased bacterial richness and diversity at different elevations have been observed as a result of increased carbon supply [54]. The significantly lower amount of available Fe and Zn in the soil of the ARC-TSC Burgershall region (84.4 and 9.6 vs. 139.7 and 17.7 mg/kg, respectively) could be another reason for the higher bacterial diversity, as the effect of high Zn and Fe in reducing bacterial diversity in agricultural soil was reported by a few studies [55,56]. Substantial differences in bacterial diversity were also observed between samples collected from the same fruit tree host, but within different geographical regions; for instance, SA3 had substantially lower bacterial diversity than SA7 (both collected from mango rhizospheres). Moreover, SA7 was collected from an orchard with a higher altitude (658 vs. 296) and soil pH (8.02 vs. 6.35), with both of these factors being reported to influence microbial diversity [52,54] (Appendix A). Furthermore, soil chemical properties such as organic matter and carbon have been reported to influence the soil microbiome and seem to also impact the diversity of soil bacteria [57]. This was the case for SA13, which had substantially higher bacterial diversity than SA6 (both collected from macadamia rhizospheres), and even though SA13 had a lower pH (5.51 vs. 8.08), it had substantially higher carbon and organic matter contents (2.48 and 4.36% vs. 0.38 and 0.66%, respectively). This is mostly due to the importance of organic carbon as the primary food source for the soil microbiome [57,58]. Therefore, it is expected that the increase in soil carbon and organic matter stimulates the diversity of the soil microbiome.

Fungal diversity among the samples collected from the ARC-TSC Nelspruit area was the highest compared to the ARC-TSC Burgershall site, which showed the lowest fungal diversity (Figure 5B). One of the main reasons for this difference is suggested to be the soil pH, as the area with the highest fungal diversity had alkaline soil, and the region with the lowest diversity had acidic soil (Table 2). Soil pH is one of the main influential factors that determine soil fungal communities [59], and increasing pH positively impacts the soil microbiome [52]. The pH influence was expected where SA9 (pH: 7.93) had a substantially higher fungal diversity than SA11 (pH: 5.77), regardless of the fact that both samples were taken from avocado rhizospheres. Higher fungal diversity in this study was associated with a region (ARC-TSC Nelspruit) with the lowest amount of K and Al and the highest amount of Fe in the soil. A positive association between soil Fe and fungal community composition was previously reported [60], because Fe facilitates the decomposition of wood for many saprotrophic fungi via the formation of hydroxyl radicals [61]. However, no information was found with regard to the possible impact of soil Al and K on the soil fungal diversity.

The microbiome extracted from SA12 (papaya rhizosphere) was the only one that resulted in a suppression of initial inoculation densities of *M. enterolobii* according to the results from the first greenhouse experiment (Figure 3). The heatmap of the top 20 bacterial (Figure 8) and fungal taxa (Figure 9) indicated that *B. aryabhattai* was the most abundant bacterial species present in this sample, while an unidentified species of the Hypocreales order was the most abundant fungal strain. The occurrence of *B. aryabhattai* could explain the highest suppressiveness of *M. enterolobii*, as this bacterial species was shown to be very effective when it comes to nematode biocontrol; previous studies proved the nematicidal potential of this bacteria against *M. javanica* and the soybean cyst nematode (*Heterodera glycines*) [62,63]. Observations from these studies suggested that *B. aryabhattai* might prevent the hatching of J2 from eggs and reduce the number of *H. glycines* cysts and *M. javanica* galls on roots, while enhancing plant growth and J2 mortality in soil [62,63]. The analyses showed the presence of several hypocrealean fungi that are known for their nematode biocontrol potential. Similar scenarios were found in some other samples of this study where the significant suppression of the nematode population was associated with the abundance of some fungal and bacterial isolates that have been reported to have nematicidal activities. These include the presence of *B. funiculus* (SA1), *B. funiculus* and *Metarhizium marquandii* (SA3), *B. simplex* and *Acremonium* sp. (SA9) and *Mortierella* sp. (SA14). The biocontrol and/or plant-growth-promoting potential of these isolates against PPNs have been recorded in previous studies [64,65,66]. No association was found for the *M. enterolobii* suppression and the abundance of particular fungal or bacterial isolates in SA2, SA6 and SA8 samples for the first experiment since the biocontrol potential of abundant genera in these sites has not been reported in the literature. The significant suppression of *M. enterolobii* by SA5 and SA13 in the repeat experiment was associated with the high abundance of fungal isolates, namely *T. rossicum* and *Talaromyces* sp., respectively. The biocontrol activity of *T. harzianum* on PPN, *Xiphinema index* in particular, has been documented [67]. Additionally, *Talaromyces flavus* has been reported to delay penetration and negatively affect the development and fecundity of *M. javanica* [68].

The beta diversity using NMDS (Figure 6 and Figure 7) indicated that SA1, SA2 and SA6 had different bacterial diversities, while SA2, SA6, SA7 and SA11 had different fungal diversities compared to other samples. There are different factors involved in shaping the soil microbial composition including geographical and environmental factors, soil physical and chemical properties as well as the host genotype [52,54,69]. It is thus anticipated that differences in microbial composition and diversity will exist between the samples collected from various geographical regions and those collected from the same location but associated with a different host [70]. The LEfSe analyses showed that 33 fungal isolates significantly differed (*p* ≤ 0.05) in terms of their abundance in the three sampling regions. These three geographical regions showed some inherent differences that could influence the fungal composition including soil pH [59], altitude and soil chemical properties, as well as climate [54,58,71].

During this study, we showed the positive role that soil microbial communities could play in reducing *M. enterolobii* population densities. Variability in terms RKN suppressiveness recorded among the treatments indicated that the suppressive potential of different soil microbiomes highly relies on the abundance and diversity of fungal and bacterial strains present in the soil. This study also revealed that the soil microbial composition is predictably influenced by several biotic and abiotic factors, and that alteration in any of these factors could shift the microbial composition. We reported that among all variables, soil dryness, pH, Fe, Zn and organic matter, altitude and crop cultivar are strong factors in shaping the soil microbiome. These observations are consistent with previous findings [25,54,55,56,59,60]. Rhizosphere microbes, especially a group of plant proactive microbes, are the first defense line that assists the host plant in protecting it during nematode invasion [21]. Consequently, increasing the abundance and diversity of such beneficial microbes is a potentially effective practice to contain severe damages caused by nematode pests. This is only practical if more sustainable approaches are considered that reduce soil disturbance and negative impacts on the soil microbiome. The results provided some insights with regard to the biocontrol potential of soil microbiomes, especially those associated with the rhizospheres of tree crops that have enough time to become well established under perennial conditions. Ultimately, this study serves as a baseline that can be used for further investigation concerning the development of microbiome-based management strategies for RKN, as these pests pose a serious threat to sustainable agricultural production and food security in sub-Saharan Africa [72].

## Figures and Tables

**Figure 1 microorganisms-10-00894-f001:**
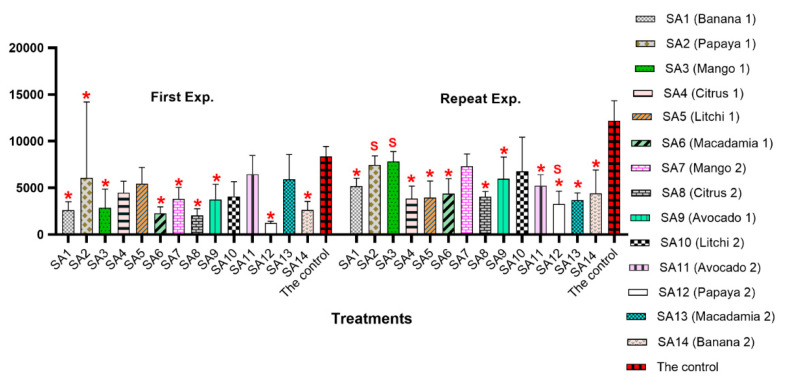
Population densities of *Meloidogyne enterolobii* in roots of tomato (Moneymaker) 7 weeks post-inoculation with 2000 eggs and second-stage juveniles and 9 weeks post-inoculation with microbial communities associated with the rhizosphere of various tropical fruit trees. Red asterisks indicate significant differences at *p* ≤ 0.05 based on Tukey’s test for each treatment compared to the control in respective experiment. Letter “S” indicates a significant difference *p* ≤ 0.05 compared to the same treatment in the first experiment.

**Figure 2 microorganisms-10-00894-f002:**
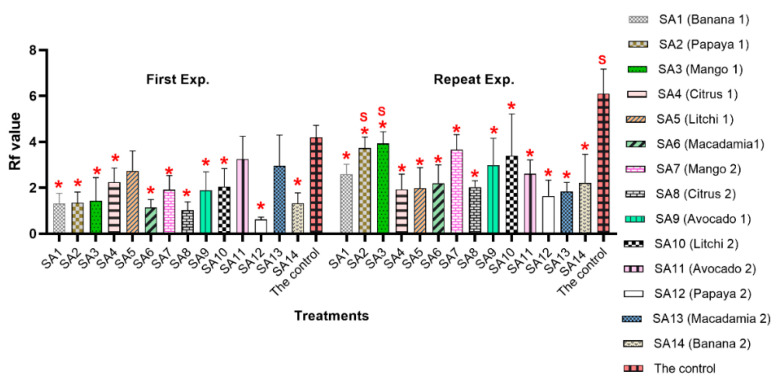
Reproduction factor (Rf) value of *Meloidogyne enterolobii* in roots of tomato (Moneymaker) inoculated with microbial communities associated with the rhizosphere of various tropical fruit trees, 7 weeks post-inoculation with 2000 eggs and second-stage juveniles. Red asterisks indicate significant differences at *p* ≤ 0.05 based on Tukey’s test for each treatment compared to the control in respective experiment. Letter “S” indicates a significant difference *p* ≤ 0.05 compared to the same treatment in the first experiment.

**Figure 3 microorganisms-10-00894-f003:**
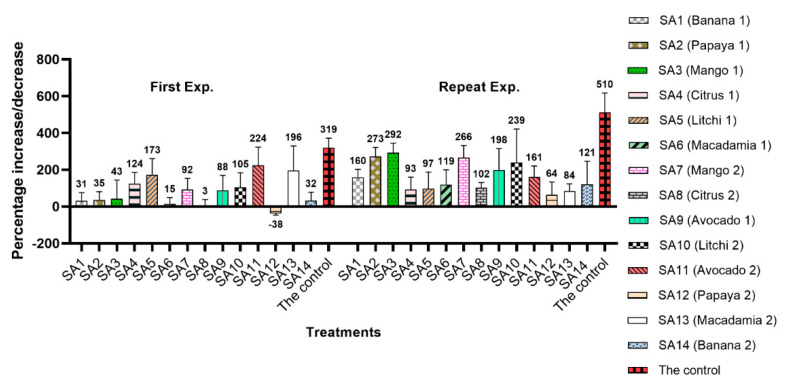
Nematode suppression effect of microbial communities associated the rhizosphere of various tropical fruit trees inoculated to roots of tomato (Moneymaker), 7 weeks post-inoculation with 2000 eggs and second-stage juveniles of *Meloidogyne enterolobii*. The *Y*-axis shows the percentage, among which, positive numbers show the increase and negative numbers indicate a reduction in the initial nematode population.

**Figure 4 microorganisms-10-00894-f004:**
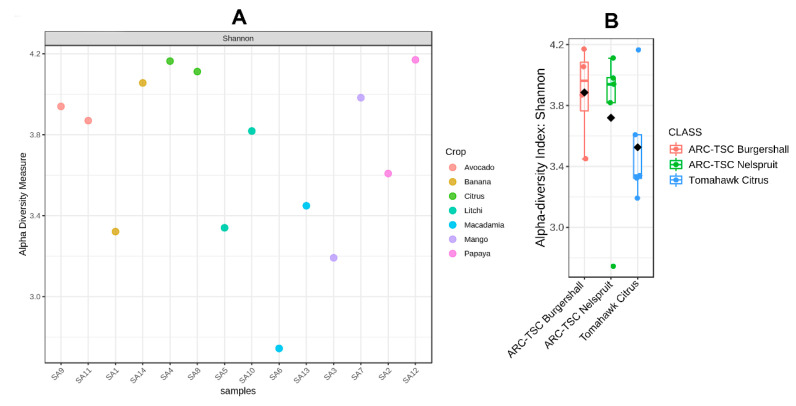
The alpha diversities of the bacterial communities associated with 14 subtropical fruit orchards in Mpumalanga province, South Africa, using the Shannon index. (**A**) Table for each sample; (**B**) alpha diversity biplot for three sampling regions.

**Figure 5 microorganisms-10-00894-f005:**
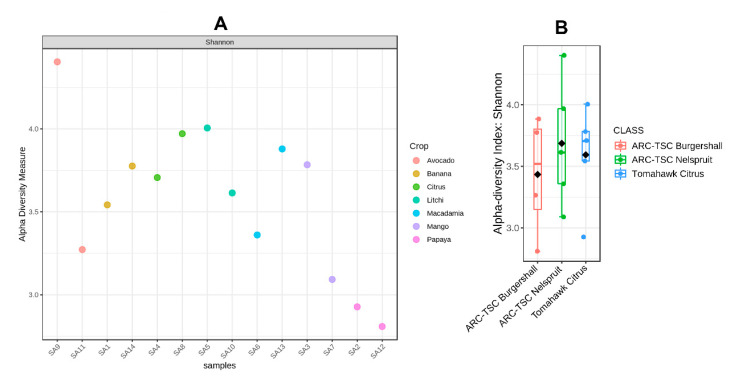
The alpha diversities of the fungal communities associated with 14 subtropical fruit orchards in Mpumalanga province, South Africa, using the Shannon index. (**A**) Table for each sample; (**B**) alpha diversity biplot for three sampling regions.

**Figure 6 microorganisms-10-00894-f006:**
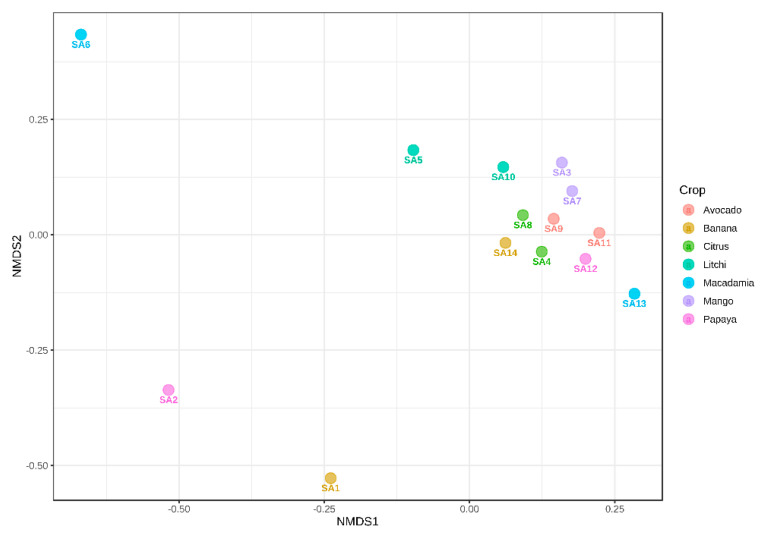
The NMDS diagram indicates the beta-diversity of bacterial communities among 14 subtropical fruit orchards from Mpumalanga province, South Africa. Permutational MANOVA (PREMANOVA) was used as the statistical method (*p* ≤ 0.81), and the Bray–Curtis dissimilarity distance was used to determine differences and similarities among the samples.

**Figure 7 microorganisms-10-00894-f007:**
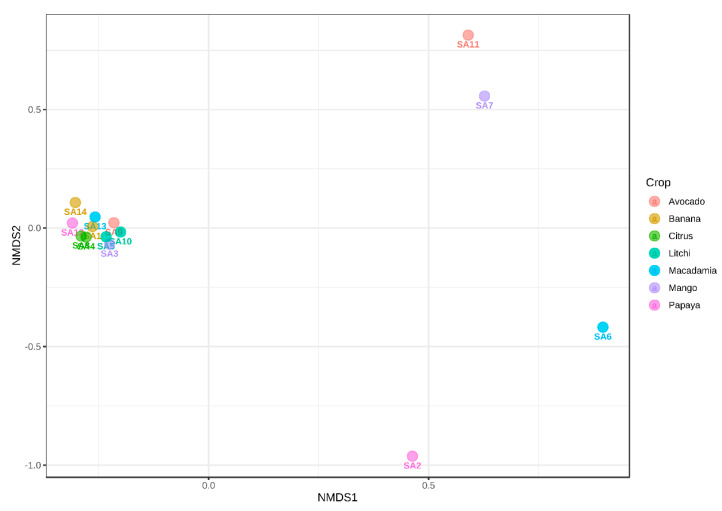
The NMDS diagram indicates the beta-diversity of fungal communities among 14 tropical fruit orchards from Mpumalanga province, South Africa. Permutational MANOVA (PREMANOVA) was used as the statistical method (*p* ≤ 0.01), and the Bray–Curtis dissimilarity distance was used to determine differences and similarities among the samples.

**Figure 8 microorganisms-10-00894-f008:**
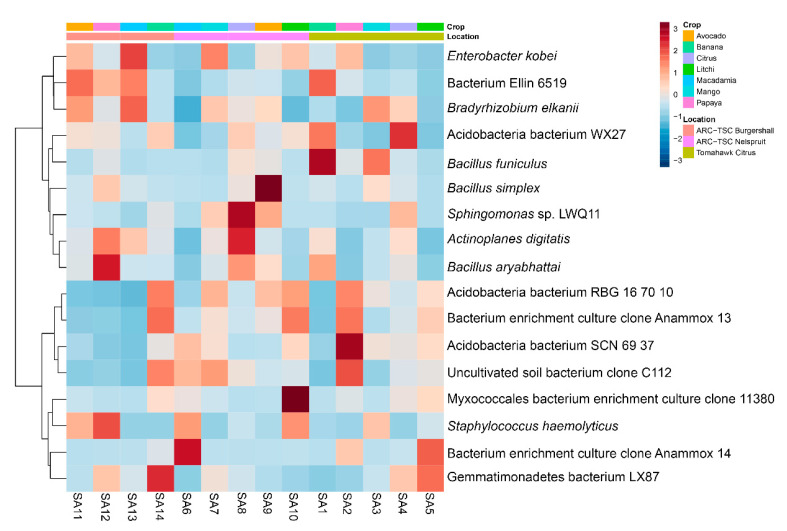
Hierarchical clustering illustrated by means of a heatmap and dendrogram for the 20 most abundant bacterial taxa associated with the rhizospheres of 14 orchards sampled in Mpumalanga province, South Africa.

**Figure 9 microorganisms-10-00894-f009:**
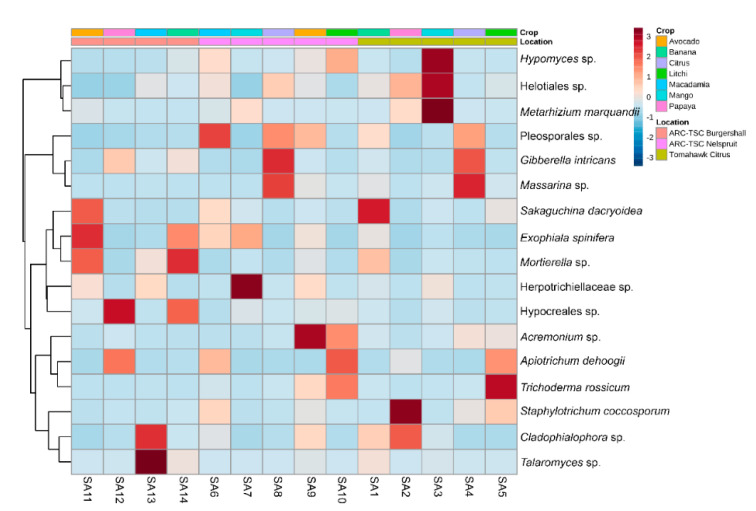
Hierarchical clustering illustrated by means of a heatmap and dendrogram for the 20 most abundant fungal taxa associated with the rhizospheres of 14 orchards sampled in Mpumalanga province, South Africa.

**Figure 10 microorganisms-10-00894-f010:**
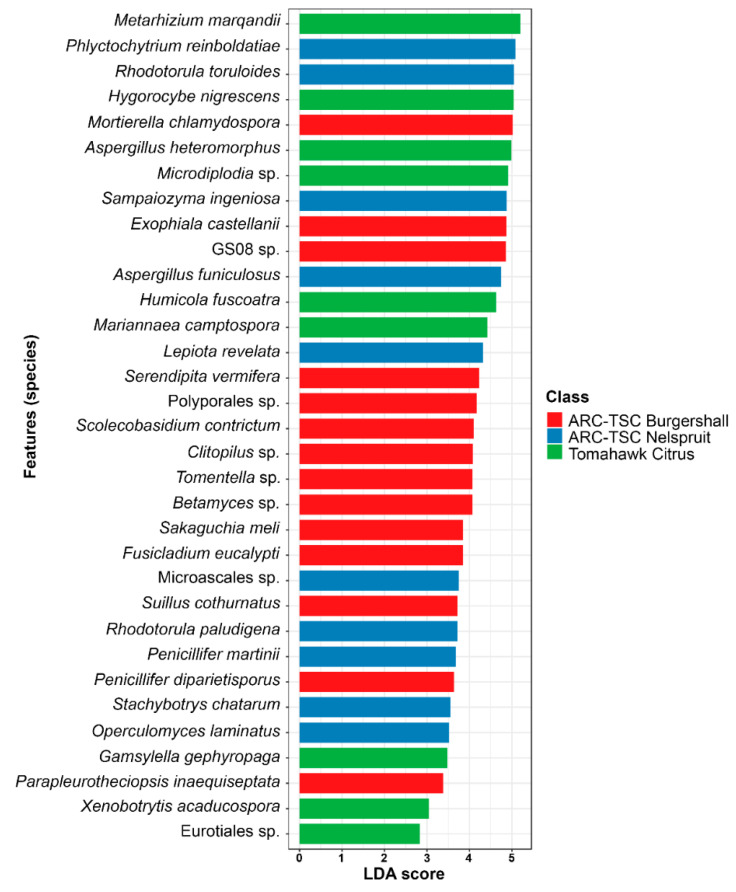
Graphical summary at fungal species level using the Linear Discriminant Analysis (LDA) Effect Size (LEfSe) based on the non-parametric factorial Kruskal–Wallis (KW) sum-rank test to identify species with significant differences (*p* ≤ 0.05) in abundance among the three sampling regions in Mpumalanga province, South Africa.

**Table 1 microorganisms-10-00894-t001:** Information regarding sampling localities and tree crops from which the soil microbial communities were extracted from rhizosphere samples to allow evaluation of their nematicidal potential against the root-knot nematode species *Meloidogyne enterolobii*.

Sample Code	Location	GPS Coordinates	Crop	Altitude (m)	Irrigation Type
SA 1	Tomahawk Citrus, Malalane	S25 36 42.3 E031 36 00.9	Banana (*Musa* sp.)	292	Microjet
SA 2	Tomahawk Citrus, Malalane	S25 37 05.7 E031 35 43.5	Papaya (*Carica papaya*)	295	Drip
SA 3	Tomahawk Citrus, Malalane	S25 37 14.2 E031 35 54.5	Mango (*Mangifera indica*)	296	Drip
SA 4	Tomahawk Citrus, Malalane	S25 37 06.8 E031 36 07.7	Citrus (*Citrus* sp.)	297	Microjet
SA 5	Tomahawk Citrus, Malalane	S25 36 38.6 E031 37 12.4	Litchi (*Litchi chinensis*)	316	Microjet
SA 6	ARC-TSC Nelspruit	S25 27 21.0 E030 58 16.1	Macadamia *(Macadamia integrifolia*)	657	Microjet
SA 7	ARC-TSC Nelspruit	S25 27 22.0 E030 58 13.8	Mango (*Mangifera indica*)	658	Microjet
SA 8	ARC-TSC Nelspruit	S25 27 22.0 E030 58 13.8	Citrus (*Citrus* sp.)	658	Microjet
SA 9	ARC-TSC Nelspruit	S25 27 24.4 E030 58 10.7	Avocado (*Persea americanum*)	654	Microjet
SA 10	ARC-TSC Nelspruit	S25 27 02.8 E030 58 21.7	Litchi (*Litchi chinensis*)	656	Microjet
SA 11	ARC-TSC Burgershall	S25 06 59.8 E031 04 56.1	Avocado (*Persea americanum*)	774	Microjet
SA 12	ARC-TSC Burgershall	S25 06 56.1 E031 04 58.1	Papaya (*Carica papaya*)	779	No irrigation
SA 13	ARC-TSC Burgershall	S25 06 57.1 E031 05 02.5	Macadamia *(Macadamia integrifolia*)	768	Microjet
SA 14	ARC-TSC Burgershall	S25 06 49.7 E031 05 08.0	Banana (*Musa* sp.)	761	Microjet

**Table 2 microorganisms-10-00894-t002:** Selected soil physical parameters as well as the altitude that differed significantly among the three geographic regions sampled. Different letters indicate significant differences for each parameter at *p* ≤ 0.05 based on Tukey’s test.

Sampling Region	pH	K mg/kg	Zn mg/kg	Fe mg/kg	Clay %	Al mg/kg	Altitude (m)
Tomahawk Citrus, Malelane	6.13 a	351.8 a	17.7 b	139.7 a	19.6 a	5.4 a	299.2 a
ARC-TSC, Nelspruit	7.98 b	56.2 b	13.09 ab	135.5 a	6.8 b	3.2 b	656.6 b
ARC-TSC Burgershall	5.94 a	348.7 a	9.6 a	80.3 b	19 a	6.7 a	770.5 c
*p* value	0.000	0.002	0.024	0.039	0.020	0.030	0.00
F ratio	34.36	10.93	5.337	4.415	5.653	4.912	5549

## Data Availability

All experimental data, including those for nematodes and soil microbes, are available and will be accessible upon request from the principal author.

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
