# Peer review of "Suppressive Effect of Soil Microbiomes Associated with Tropical Fruit Trees on Meloidogyne enterolobii"

_microorganisms, 2022, doi:10.3390/microorganisms10050894_

Round 1

Reviewer 1 Report

The manuscript is well written and the results support the conclusion. The authors have done a thorough research & analysis. They have also bolstered their findings with additional experiments.      I recommend that the manuscript be accepted after minor revision.   Line numbers missing. Makes it difficult to pinpoint issues. Please follow the mdpi guidelines.   It is not mentioned anywhere in the introduction why the microbial suppressive effect was tested only on M. enterolobii? Can the authors elaborate on this point?    The authors have mentioned the term "genotype" at several instances in the manuscript. I am not sure what the authors are referring to with respect to genotype? I do not see a genotypic analysis of cultivars in this study.    Pg.2 - Reference format for Kishi et al not in numbers. Please rectify. Also, Please check out some recent papers such as "Managing nematodes without methyl bromide" by Zasada et al 2010  or "Phytochemical based strategies for nematode control" by Chitwood in Annual reviews of Phytopathology.   ​Formatting issues with references - Some years are bold, unnecessary space between some citations.​

Author Response

The manuscript is well written and the results support the conclusion. The authors have done a thorough research & analysis. They have also bolstered their findings with additional experiments. I recommend that the manuscript be accepted after minor revision.   Line numbers missing. Makes it difficult to pinpoint issues. Please follow the mdpi guidelines.   

The authors thank the reviewer for the compliment and also would like to apologize for the inconvenience due to the missing line number.

It is not mentioned anywhere in the introduction why the microbial suppressive effect was tested only on M. enterolobii? Can the authors elaborate on this point? 

The following info was added to the introduction   

“Therefore, this study aimed to evaluate the nematode control potential of the soil microbiomes associated with the rhizospheres of various perennial subtropical tree crops on Meloidogyne enterolobii, widely spread species in subtropical regions of South Africa [9], under greenhouse conditions.”

The authors have mentioned the term "genotype" at several instances in the manuscript. I am not sure what the authors are referring to with respect to genotype? I do not see a genotypic analysis of cultivars in this study. 

The word genotype has been used to refer to the cultivars and since the authors did not perform any genotype analysis all genotype words were replaced with cultivar in the Manuscript.

Pg.2 - Reference format for Kishi et al not in numbers. Please rectify. Also, Please check out some recent papers such as "Managing nematodes without methyl bromide" by Zasada et al 2010  or "Phytochemical based strategies for nematode control" by Chitwood in Annual reviews of Phytopathology.  â€‹

The reference has been omitted from the text as the reviewer asked.

Formatting issues with references - Some years are bold, unnecessary space between some citations.​

Thank you for pinpointing this, the authors tried to correct all issues with respect to the citation.

Reviewer 2 Report

The authors of the reviewed manuscript presented the results of their study on the effectiveness of soil microbiomes, associated with various subtropical fruit trees, in managing the population of the harmful species of  Meloidogyne enterolobii. The  obtained results are well documented and discussed.

The conducted research expanded the existing knowledge on the soil and plant microbiomes and their interactions with pests.

I recommend the article for print after minor revision.

Suggested corrections:

- page 2, line6 - the publication cited here (Kishi et al. 1995) is not included in References.

 - page 6, line 5 - please add the citation number [44]

- page 9, in 3.5 Alpha diversity, line 3 "SA12 from citrus" -  there is some mistake here (?). It should be probably "SA12 from papaya and SA4 from citrus"

- page 20, 6 line from bottom - is 670 it should be 70

Author Response

- page 2, line6 - the publication cited here (Kishi et al. 1995) is not included in References.

The reference has been omitted from the text as the first reviewer asked.

 - page 6, line 5 - please add the citation number [44]

Corrected in the text.

- page 9, in 3.5 Alpha diversity, line 3 "SA12 from citrus" -  there is some mistake here (?). It should be probably "SA12 from papaya and SA4 from citrus"

Was corrected as “SA12 and SA4 from papaya and citrus rhizospheres, respectively”

- page 20, 6 line from bottom - is 670 it should be 70

Corrected

This manuscript is a resubmission of an earlier submission. The following is a list of the peer review reports and author responses from that submission.

Round 1

Reviewer 1 Report

This manuscript describes the suppressive effects of soil microbiomes associated with tropical fruit trees on Meloidogyne enteromobii, indicating that the nematode suppression potential of different soil microbiomes highly depends on the abundance and diversity of fungal and bacterial strains present in soil. The manuscript was written well and the results may be quite attractive to the readers.

I think this manuscript can be published in this journal. However, the authors should address the following questions before acceptance.

  1. I think that the number of 2,000 eggs and J2s were too low to cause galls on the tomato roots. Please provide the gall index and egg mass of each treatment 6 weeks after inoculation.
  2. Did you count the number of J2s in the soil samples where the tomato plants were grown?
  3. Did you repeat the greenhouse experiments?
  4. “The 4-leaf stage tomato seedlings were planted in the compost mixture were transplanted into the pots with each pot receiving one tomato plant.‘ --> Rewrite
  5. “The NMDS based on the fungal dataset indicated the following orchards: S11, SA7, SA6 and SA2 grouped“ --> SA11??

Reviewer 2 Report

The study focused on the soil microbiome associated with several fruit and nut trees in 14 orchards. The authors hypothesized that some of these soil biomes might reduce tomato crop damage by Meloidogyne enterolobii

The topic of microbial protection against plant-parasitic nematodes has become increasingly popular with the development of next-generation sequencing. It follows earlier publications by trying to mitigate crop damage or reduce the population density of root-knot nematodes after introducing potentially suppressive rhizobiome compared to an untreated control. Still, the manuscript is timely and original. It is within the scope of the journal and should be of interest to a broad readership.

However, for the bioassay, a single tomato trial was performed. The suppressiveness study and following microbial analysis are based on a non-repeated experiment. The whole research is therefore lacking evidence of reproducibility.